# Assessing Smoking Status and Risk of SARS-CoV-2 Infection: A Machine Learning Approach among Veterans

**DOI:** 10.3390/healthcare10071244

**Published:** 2022-07-04

**Authors:** Alice B. S. Nono Djotsa, Drew A. Helmer, Catherine Park, Kristine E. Lynch, Amir Sharafkhaneh, Aanand D. Naik, Javad Razjouyan, Christopher I. Amos

**Affiliations:** 1VA HSR&D Center for Innovations in Quality, Effectiveness and Safety, Michael E. DeBakey VA Medical Center, Houston, TX 77030, USA; alice.nonodjotsa@va.gov (A.B.S.N.D.); drew.helmer@va.gov (D.A.H.); catherine.park@va.gov (C.P.); amirs@bcm.edu (A.S.); anaik@bcm.edu (A.D.N.); 2Department of Medicine, Baylor College of Medicine, Houston, TX 77030, USA; chris.amos@bcm.edu; 3Big Data Scientist Training Enhancement Program (BD-STEP), VA Office of Research and Development, Washington, DC 20571, USA; 4VA Salt Lake City Health Care System, University of Utah, Salt Lake City, UT 84148, USA; kristine.lynch@va.gov; 5Department of Management, Policy and Community Health, UTHealth School of Public Health, Houston, TX 77030, USA

**Keywords:** SARS Coronavirus 2, smoking, machine learning, veteran

## Abstract

The role of smoking in the risk of SARS-CoV-2 infection is unclear. We used a retrospective cohort design to study data from veterans’ Electronic Medical Record to assess the impact of smoking on the risk of SARS-CoV-2 infection. Veterans tested for the SARS-CoV-2 virus from 02/01/2020 to 02/28/2021 were classified as: Never Smokers (NS), Former Smokers (FS), and Current Smokers (CS). We report the adjusted odds ratios (aOR) for potential confounders obtained from a cascade machine learning algorithm. We found a 19.6% positivity rate among 1,176,306 veterans tested for SARS-CoV-2 infection. The positivity proportion among NS (22.0%) was higher compared with FS (19.2%) and CS (11.5%). The adjusted odds of testing positive for CS (aOR:0.51; 95%CI: 0.50, 0.52) and FS (aOR:0.89; 95%CI:0.88, 0.90) were significantly lower compared with NS. Four pre-existing conditions, including dementia, lower respiratory infections, pneumonia, and septic shock, were associated with a higher risk of testing positive, whereas the use of the decongestant drug phenylephrine or having a history of cancer were associated with a lower risk. CS and FS compared with NS had lower risks of testing positive for SARS-CoV-2. These findings highlight our evolving understanding of the role of smoking status on the risk of SARS-CoV-2 infection.

## 1. Introduction

As of 1 July 2022, approximately 548 million cases and approximately 6.3 million deaths worldwide have been attributed to coronavirus 2 (SARS-CoV-2) infections [1]. Our current understanding is that COVID-19 is a multifactorial disease caused by the SARS-CoV-2 virus, in which infection and morbidity depend on multiple factors [2]. Smoking likely plays an important role in the COVID-19 pandemic, but what that role is exactly remains unclear.

Smoking is a huge healthcare concern because it is common and an established major risk factor for most preventable diseases, including certain types of cancers, cardiovascular diseases, diabetes, respiratory illnesses, and other infectious diseases [3,4,5]. Many of these conditions, such as respiratory illnesses [2,6,7,8], cardiovascular disease [2,7], cancer [7], chronic kidney disease [6,7], chronic diabetes [6,7], obesity [6], hypertension [7], and neurological diseases [2], are also important risk factors for worse clinical outcomes (hospitalization, ICU admission, and death) in patients with COVID-19. 

Studies have reported either reduced risks of testing positive for SARS-CoV-2 among Current or Former Smokers [9,10,11], no difference in risk of infection among Current and Former Smokers [10], and even an increased risk among Former or Current Smokers compared with Never Smokers [12]. However, limitations of these studies include small sample sizes, assessing few comorbidities, being conducted early in the pandemic, or lacking detailed smoking status. A better understanding of the actual link between smoking status and risk of COVID infection and adverse outcomes is important to mitigate those risks and optimize outcomes.

The Veteran’s Hospital Administration (VHA) has compiled data on COVID-19-related healthcare for veterans from across the US since 2020. It offers a large cohort with reliable smoking status (Current, Former, and Never Smokers), demographic characteristics, pre-existing comorbidities, and pre-infection medications. This study assessed the association between smoking status and testing positive for SARS-CoV-2 infection in a large population of US military veterans who use the nation’s largest integrated health system. We hypothesized that there are differences in risk of testing positive for SARS-CoV-2 infection between Current and Former Smokers compared with Never Smokers.

## 2. Materials and Methods

### 2.1. Study Design, Participants, and Data Source

We conducted a retrospective cohort study to investigate the impact of smoking on testing positive for SARS-CoV-2 infection using the VA COVID-19 Shared Data Resource, established to collect COVID-19 infection data and facilitate research. This domain contains specific, curated information for patients seen at VHA medical centers and tested for SARS-CoV-2 infection since February 2020 [13]. Patients were eligible if they were (1) unvaccinated, (2) tested for SARS-CoV-2 infection between 1 February 2020 and 28 February 2021, and (3) had at least one encounter (inpatient or outpatient visit) at a VHA facility over 2 years, from January 2019 to December 2020. Patients were excluded if they did not have at least one encounter during that period. We identified 1,176,306 veterans who were tested for SARS-CoV-2 infection that met the inclusion criteria (Figure 1). We defined the index date as the date of the first positive or negative SARS-CoV-2 test. The study was approved by the Baylor College of Medicine Institutional Review Board (H-47595) and the Michael E. DeBakey VA Medical Center Research and Development Committee.

### 2.2. SARS-CoV-2 Test Positivity, Smoking, and Other Variables

The primary outcome of the SARS-CoV-2 test positivity was determined using a reverse transcription polymerase chain reaction (RT-PCR) SARS-CoV-2 test. We used the first date of a positive test to construct the cohort and, therefore, included only the first positive test result. The main exposure, smoking status, was obtained from the VHA Electronic Medical Record (EMR) Health Factors (HF) dataset. The HF table contains longitudinal data on patients routinely generated by clinical visits and stored within the EMR, including smoking status and history. The smoking status data were mapped to distinct categories based on the most recent updates (Current, Former, Never, and Unknown Smokers) from the HF dataset. Prior data were used to resolve any discrepancies. Specifically, any Current or Former Smoker who became a Never Smoker was identified as a Former Smoker. Prior research reported a high agreement between records in the EMR and self-reported smoking status gathered from questionnaires with reported kappa statistics ranging from 0.66–0.74 [14,15,16]. We also included demographic variables (age), clinical characteristics (BMI), pre-existing conditions (hypertension, cancer, diabetes mellitus, etc.), and pre-infection medications. Patients’ smoking status, pre-existing conditions, and pre-infection medications were obtained for 2 years prior to the index date.

### 2.3. Data Processing and Machine Learning

To facilitate the identification of the most important variables during the variable selection process, continuous variables were transformed to categorical variables (age: ≤30, 30–40, 40–50, 50–65, 65–75, 75–85, and ≥85 years; body mass index (BMI): <18.5, 18.5–30, and ≥30 kg/m^2^).

#### 2.3.1. Imputation Process

There were missing data for smoking status; 10.0% for all veterans and 10.7% for those who tested positive. We used the Multiple Imputations by Chained Equations in R [17] to impute the unknown smoking status data to Current, Former, or Never and the missing data for age and BMI. To increase the generalizability of the imputation model, all available covariates without missingness were included, and five datasets were imputed using five iterations.

#### 2.3.2. Identifying the Most Important Variables Using a Machine Learning Selection Process

We illustrate the process of the most important variable selection in Figure 2, as previously described [6]. We sequentially used a cascade of machine learning approaches in four steps to identify the most important predictors for a positive SARS-CoV-2 test using both imputed and unimputed datasets. The variables were curated from the full breadth of EMR (i.e., demographic, clinical, pre-existing conditions, and pre-infection medications). We started with 165 initial variables. First, we removed any variable with a prevalence of less than 1%, and this step reduced the dimensionality to 119 variables. Second, a univariate filter method (a chi-squared test used to evaluate the significance of each independent variable to the target variable) excluded variables not statistically associated with testing positive for SARS-CoV-2 infection at *p* < 0.05. We retained 108 variables. Third, we applied an embedded method, the least absolute shrinkage and selection operator (LASSO) [18], with 10-fold cross-validation to select the most important variables; this step kept 76 variables. LASSO is a regression model that adds shrinkage (regularization penalty) to shrink the coefficients of less contributory variables to zero. The 10-fold cross-validation split the dataset into ten equal parts. Nine parts were used for training and one part for validation. The process was repeated ten times. The variables that were retained most frequently in predictive models for SARS-CoV-2 infection were selected as the most important variables. Finally, we applied a wrapper method, the sequential forward selection (SFS) variable with a five-fold cross-validation. This final step selected 12 variables. SFS starts with an empty set of variables, then it identifies the best variable (associated with the best performing single regression model based on a *p*-value selection criterion); next, it evaluates all possible models (pairs) of the best variable and each of the remaining variables and selects the best pair. It sequentially adds a new variable to the preselected variables until the variable addition does not reduce the criterion-value by less than 0.05. The full lists and set of variables selected at each step are presented in Appendix A. The analysis of unimputed data after removing patients with unknown smoking status provided an identical list of features at each step.

### 2.4. Statistics

Descriptive analyses were conducted to summarize the cohort characteristics by smoking status using the unimputed data. Mean and standard deviation (±SD) were reported for continuous variables; frequency count (N) and percentage (%) were computed for categorical variables; and Chi-square statistics and ANOVA (*p*-values) evaluated the difference between the groups of smokers. The associations between smoking status and a positive SARS-CoV-2 test were assessed using two binary logistic regression models to estimate the odds ratios (OR) and report 95% confidence intervals (95% CI). The first model adjusted for the most important covariates identified through the feature selection process described above (age (older age), BMI (being overweight and being underweight), cancer, dementia, Hispanic or Latino, lower respiratory infections, pneumonia, sex (male), septic shock, and current smoking). The second model used the same covariates but stratified the patients by age (age <65 versus ≥65). The imputed dataset was used for primary analysis and the unimputed dataset for sensitivity analysis. Statistical analyses and machine learning were performed using Python 3.8.3.

## 3. Results

### 3.1. Participant Characteristics Overall and by Smoking Status

The 1,176,306 veterans who were tested for SARS-CoV-2 infection at the VHA during the study period had a positivity rate of 19.6% (Table 1). They were predominantly men (86.5%), white (65.6%), and non-Hispanic or not Latino (84.9%). They had a mean age of 60.4 ± 16.0 years and BMI of 30.0 ± 6.2 kg/m^2^. Of these patients, 380,648 (32.3%) were Never Smokers; 456,348 (38.8%) were Former Smokers; 221,515 (18.8%) were Current Smokers; and 117,795 (10.0%) had an Unknown smoking status (Table 1, Appendix A). The imputation of Unknown smoking status changed the number of cases in the Never Smoker category from 380,648 to 432,078 (+4.4%), Former Smoker from 456,348 to 506,206 (+4.2%), and Current Smoker from 221,515 to 230,022 (+0.8%) (Table 2, Appendix A).

Among the 230,250 (19.6%) veterans who tested positive for SARS-CoV-2, 85,605 (37.2%) were Never Smokers; 93,665 (40.7%) were Former Smokers; 26,809 (11.6%) were Current Smokers; and 24,171 (10.5%) had an Unknown smoking status (Figure 1, Table 2). The imputation of Unknown smoking status changed the number of cases in the Never Smoker category from 85,605 to 97,419 (+5.1%), Former Smoker from 93,665 to 103,482 (+4.3%), and Current Smoker from 26,809 to 29,349 (+1.1%) (Table 2, Appendix A).

### 3.2. Most Important Variables

The most important variable selection process started with 165 initial features and ended with the 12 most important variables (Figure 2). Among these 12 most important variables, four were protective, including Current Smoker status (aOR: 0.51; 95% CI: 0.50, 0.52; *p* < 0.0001), phenylephrine (aOR 0.76; 95% CI: 0.73, 0.79; *p* < 0.0001), cancer (aOR 0.72; 95% CI: 71, 0.73; *p* < 0.0001), and being underweight (BMI < 18.5 (aOR 0.73; 95% CI: 0.70, 0.77; *p* < 0.0001)). The remaining features were associated with an increased risk of testing positive for SARS-CoV-2: older age (age ≥ 85 (aOR 1.33; 95% CI: 1.30, 1.36; *p* < 0.0001)), being overweight (BMI ≥30; aOR 1.31; 95% CI: 1.30, 1.32; *p* < 0.0001)), sex (Male; aOR 1.18; 95% CI: 1.16, 1.19; *p* < 0.0001), being Hispanic or Latino (aOR 1.17; 95% CI: 1.16, 1.20; *p* < 0.0001), dementia (aOR 1.42; 95% CI: 1.39, 1.45; *p* < 0.0001), lower respiratory infections (aOR 1.09; 95% CI: 1.07, 1.11; *p* < 0.0001), pneumonia (aOR 1.20; 95% CI: 1.17, 1.22; *p* < 0.0001), and septic shock (aOR 1.07; 95% CI: 1.04, 1.10; *p* < 0.0001) (Table 3, Appendix A).

### 3.3. SARS-CoV-2 Test Positivity and Pre-Existing Respiratory Illnesses

Among the 15 pre-existing respiratory illnesses in our dataset, the most prevalent were chronic lung disease, obstructive sleep apnea, chronic obstructive pulmonary disease, lower respiratory infection, bronchitis, asthma, pneumonia, and acute respiratory failure (Appendix A). Ten of these variables were selected using univariate analysis, the LASSO step kept six, and only two remained after the final step as important risk factors: lower respiratory infections (aOR 1.09; 95% CI: 1.07, 1.11; *p* < 0.0001) and pneumonia (aOR 1.20; 95% CI: 1.17, 1.22; *p* < 0.0001) (Table 3, Appendix A).

### 3.4. SARS-CoV-2 Test Positivity and Smoking Status

In the univariate analysis, the unadjusted odds of testing positive for SARS-CoV-2 infection were significantly lower in Current Smokers (OR: 0.48; 95% CI: 0.47, 0.49; *p* < 0.0001) and Former Smokers (OR: 0.88; 95% CI: 0.87, 0.89; *p* < 0.0001) compared with Never Smokers (Table 3 and Table 4), using imputed and unimputed data. In multivariate modelling, relative to Never Smokers, the odds of testing positive remained lower after adjusting for the 12 most important variables in Current Smokers (aOR: 0.51; 95% CI: 0.50, 0.52; *p* < 0.0001) and in Former Smokers (aOR: 0.89; 95% CI: 0.88, 0.90; *p* < 0.0001) (Table 3 and Table 4).

### 3.5. SARS-CoV-2 Test Positivity and Smoking Status Stratified by Age

To address the marked difference in the risk between younger and older veterans, veterans were stratified by age (age <65 versus age ≥65), and risk estimates were adjusted for the top 12 variables, except for age ≥85 (Table 4, imputed data; Appendix A, imputed and unimputed data). Among veterans aged ≥65 years, the Current Smoker status only was associated with a lower risk of testing positive in unadjusted (OR: 0.50; 95% CI: 0.49, 0.51; *p* < 0.0001) and adjusted odds (aOR: 0.52; 95%CI: 0.51, 0.53; *p* < 0.0001). Former Smoker status was inconclusive in older veterans with unadjusted odds (OR: 0.99; 95% CI: 0.97, 1.00; *p* > 0.05) and adjusted odds (aOR: 0.98; 95%CI: 0.97, 1.00; *p* > 0.05). For veterans aged < 65 years, Current Smokers had a decreased risk reflected by unadjusted (OR: 0.47; 95% CI: 0.46–0.48; *p* < 0.0001) and adjusted odds (aOR: 0.48; 95% CI: 0.47, 0.49; *p* < 0.0001). Testing positive remained significantly lower based on Former Smoker status and age < 65 years, indicating it was protective based on unadjusted (OR: 0.83; 95% CI: 0.81, 0.84; *p* < 0.0001) and adjusted odds (aOR: 0.83; 95%CI: 0.81, 0.84; *p* < 0.0001). Very similar associations were observed when using unimputed data (Table 4). 

## 4. Discussion

We analyzed data from a large cohort and found that currently smoking is associated with a lower risk of testing positive among veterans tested for SARS-CoV-19 at VHA facilities across the United States. Relative to Never Smokers, the risk of testing positive was lower in Current Smokers (aOR: 0.51; 95%CI: 0.50, 0.52; *p* < 0.0001) and Former Smokers (aOR: 0.89; 95%CI: 0.88, 0.90; *p* < 0.0001) after adjusting for the 12 most important variables selected using machine learning techniques. Furthermore, an age-stratified analysis showed that, among younger veterans, being a Current Smoker or a Former Smoker was protective, while in older veterans, a decreased risk of SARS-CoV-2 infection was present only in Current Smokers.

Why current smoking is associated with a lower risk of testing positive for SARS-CoV-2 remains unclear. Previous epidemiologic studies have reported findings of an apparent protective effect from testing positive for SARS-CoV-2 among smokers, although some studies were not actually looking for the effect of smoking status. In our analysis, smoking was the main variable of interest. Our findings extend previously reported findings of lower risks of testing positive for SARS-CoV-2 in Current Smokers (OR: 0.52) and Former Smokers (OR: 0.92) in the same population of veterans using a smaller cohort (n = 88,747) tested early in the pandemic [9]. Our results are also consistent with a decreased risk of SARS-CoV-2 infection (hazard ratio range 0.40 to 0.48) observed among heavy, moderate, and light smokers in a cohort study of 8.28 million participants, 19,486 of whom tested positive [19].

A systematic review of 233 studies [20] reported a lower prevalence of smokers among those individuals who tested positive. A subsequent meta-analysis of studies with detailed smoking status (Current, Former, and Never Smokers) also demonstrated a reduced risk of infection in Current Smokers (RR: 0.74) compared with Never Smokers. Despite the number of studies available, only one was rated with good quality. Jose et al. [21] described the risk in 69,264 patients, including reduced risks for testing positive in patients who smoked cigarettes only and patients who both smoked and vaped relative to non-smoker/non-vapers but similar risk among those who vaped only. These findings suggest the lower risk of testing positive for SARS-CoV-2 is unique to cigarette smoking. In a cohort of 22,914 veterans with cancer, Current Smokers had a significantly lower prevalence (5.3%) for positive SARS-CoV-2 tests compared with Former and Never Smokers combined (9.5%) [22].

By contrast, a recent study triangulating observational analyses (OA) and Mendelian randomization (MR) reported that former smoking (37%) increased the risks of infection, hospitalization, and death in OA; current smoking (3%) only increased the risks of hospitalization and death; in MR, both smoking initiation and heaviness increased the risks of all three outcomes (OR range 1.45 to 10.02), indicative of a causal effect of smoking on COVID-19 severity [23]. Likewise, data from two survey-based studies documented higher risks of infection among Current Smokers [24,25]. Further, two meta-analyses of hospital-based studies [20,26] revealed that Current and Former Smokers are more likely to experience severe COVID-19 complications, such as hospitalization, disease severity requiring ICU, and death. Our previous study found that Former Smokers had an increased risk of in-hospital mortality, whereas the risk was similar between Current and Never Smokers [6].

Testing positive for SARS-CoV-2 infection was also associated with sociodemographic and pre-infection clinical factors. Older age, being overweight (BMI ≥30 kg/m^2^), being Hispanic or Latino, and male sex were associated with increased risk of testing positive. Other studies corroborated the increased risks of testing positive attributable to older age [9,12,27,28], male sex [9,12,28], being overweight [9,12,28,29], and being Hispanic or Latino [9,27,28,30]. Similarly, four pre-existing conditions were associated with an increased risk of testing positive for SARS-CoV-2 infection in our cohort, including dementia (aOR: 1.42; 95% CI: 1.39, 1.45), lower respiratory infection (aOR 1.09; 95% CI: 1.07, 1.11), pneumonia (aOR 1.20; 95% CI: 1.17, 1.22), and septic shock (aOR 1.07; 95% CI: 1.04, 1.10). Similar associations were reported in other cohorts [28,31,32,33]. These individual factors associated with testing positive for SARS-CoV-19 should inform public health risk stratification and mitigation interventions and messages.

Two other individual factors were associated with reduced risk of testing positive. Patients with a history of cancer had a significantly reduced risk (aOR 0.72; 95% CI: 71, 0.73). Previous studies also reported a reduced risk of infection among cancer patients [7,11]. It is possible that cancer patients may be more likely to get tested when asymptomatic; thus, the reduced risk may be related to care seeking behaviors and screening for COVID-19. People who had a past prescription of phenylephrine before testing were also at lower risk (aOR 0.76; 95% CI: 0.73, 0.79) of testing positive for SARS-CoV-19. This is a decongestant medication generally prescribed to treat stuffy nose, cough pain, and fever. It is possible that people with regular upper respiratory symptoms who use this decongestant may be more likely to get tested than other people.

Unlike previous epidemiological studies that rely on a predetermined set of covariates, our larger sample size permitted application of a machine learning approach to take full advantage of each individual variable in the EMR. This explains why our model identified pre-existing conditions and pre-infection medications as independent factors associated with positive SARS-CoV-19 test results, despite their relatively low frequency in the sample. Other known comorbidities and risk factors for severe COVID-19 [34,35,36,37], such as diabetes, cardiovascular disease, hypertension, and chronic kidney disease, were selected at intermediate steps during the variable selection process but were not retained as the most important variables in the last sequential forward step.

Knowledge about physiological mechanisms that could confer lower risk for SARS-CoV-2 infection among smokers remains underdeveloped. Prior studies by us [38] and others [39,40] have shown that smoking upregulates ACE-2 receptors in lung tissue, particularly in Goblet cells [38], which could make Current Smokers more susceptible to lung infections from SARS-CoV-2, which is not what we found in this study. Nicotine exposure upregulates many subtypes of nicotinic acetyl choline receptors. Upregulation of the α7 pentamer could have an anti-inflammatory effect, whereas α3β4 pentamer upregulation increases particle transport speed, which could help flush pathogens. Studies of vapers, who are exposed to nicotine but avoid many toxic substances in tobacco cigarettes, have failed to demonstrate altered risks of SARS-CoV-2 infection according to vaping status [41], which argues against nicotine itself playing a substantial role [42] in protecting smokers for SARS-CoV-2 infections. Similarly, a clinical trial evaluating nicotine application in hospitalized COVID-19 patients showed no benefit [43]. An alternate hypothesis is that upregulation of the RAS pathway due to toxicants in cigarette smoke eventually leads to the apoptosis of alveolar epithelial cells and the proliferation of fibroblasts, reducing targets for COVID-19. One RAS pathway member upregulated in smokers, Angiotensin 1–7, has anti-proliferative and anti-inflammatory activities, unlike other RAS members that are proinflammatory. A homeostatic balance between inflammatory and anti-inflammatory components of the RAS pathway could influence protection from COVID-19 among smokers, but further studies are needed to understand the specific protection from COVID-19 that seems to be provided to Current Smokers but not Former Smokers [44].

### Strengths and Weaknesses

Recent critiques have argued that studies reporting an inverse correlation between smoking and testing positive have several methodological flaws, such as not having smoking as the primary exposure, low prevalence of smokers in cohorts, lacking explicit data on smoking status, not adjusting for confounders associated with smoking, and unexpectedly low percentages of chronic obstructive pulmonary disease and cardiovascular disease particularly among smokers, indicating potential selection bias [45,46]. Veterans are known to use tobacco products at a higher rate than non-veterans [47,48,49]. According to a 2016 report, 32.5% of veterans who served after 2001 are Current Smokers, 24.8% are Former Smokers, and 42.7% are Never Smokers [47]. A recent report described a decline in the proportion of VHA enrollees who are Current Smokers (15.9% in 2017 to 12.9% in 2021); now 41.9% are Former Smokers, and 43.9% are Never Smokers [50]. Our large cohort has an appropriate representation of Current (18.8%) and Former Smokers (38.8%), with the prevalence of Current Smokers above the rate in VHA over the last 5 years. Similarly, cardiovascular disease (all 35.7%, Never Smoker 29.7%, Former 41.6%, Current Smoker 33.7%) and chronic obstructive pulmonary disease (all 19.1%, Never Smoker 9.5%, Former 22.3%, Current Smoker 28.9%) had appropriate representation of smokers, reducing concerns about bias.

Major strengths of this analysis include the large cohort with a national scope assessing the association between smoking status and testing positive for SARS-CoV-2. The VHA is known to have a high agreement of coding smoking status in EMR. Notably, we imputed missing smoking status and obtained similar results between unimputed and imputed datasets. To avoid confounding with vaccination, we restricted our analysis to the period prior to widespread vaccination. We used machine learning to select the most important variables we adjusted for in multivariate analysis. In future studies, application of a broader suite of machine learning tools could identify additional factors and interactions among them that predict risk for COVID-19 infections [51]. A recent review of applications of many different machine learning applications found that the regularized logistic regression we used is widely used and has a comparable accuracy to many other applications [52].

Our study has some limitations. The study population is veterans, mostly men seeking care in the VHA, with a higher burden of comorbidities compared to the general US population. There is potential for selection bias; some of our findings were likely affected by differences in accessing care at the VHA. It is possible that smokers who are sick and tested positive were getting their care outside the VHA, while the fewer sick smokers were coming to the VHA and less likely to test positive. These socially determined behaviors could partially explain the discrepant association between smoking and contracting SARS-CoV-19. However, in an age-stratified analysis to create subgroups of more similar healthcare seeking behaviors, the protective effects of current smoking were comparable in older and younger veterans. The lack of association between a positive SARS-CoV-19 test and Former Smokers among older veterans could reflect a longer period of abstention in older patients; thus, their risk becomes similar to Never Smokers.

## 5. Conclusions

This study suggests that currently smoking is associated with a decreased risk for testing positive for SARS-CoV-2 among veterans receiving their testing at VHA facilities. Further research should continue to explore the relationship between smoking and SARS-CoV-2 infection and outcomes to develop and deliver clear messages about risk and harm mitigation.

## Figures and Tables

**Figure 1 healthcare-10-01244-f001:**
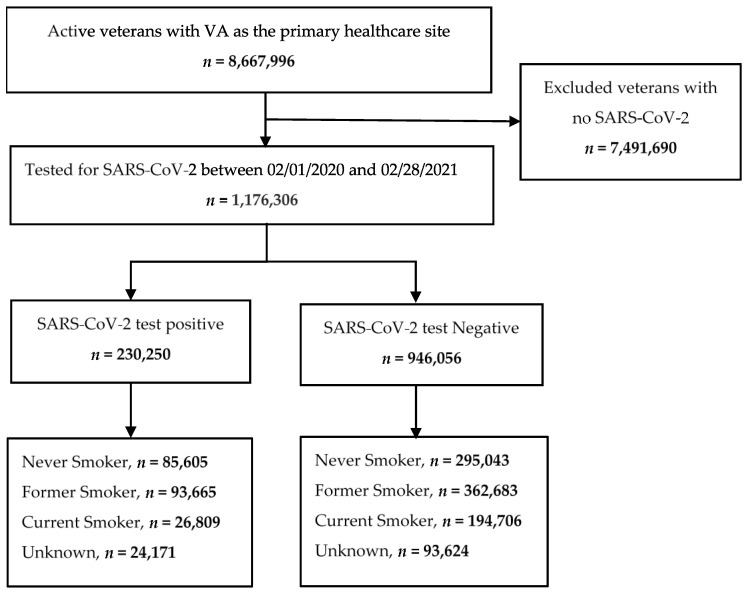
Strobe diagram.

**Figure 2 healthcare-10-01244-f002:**
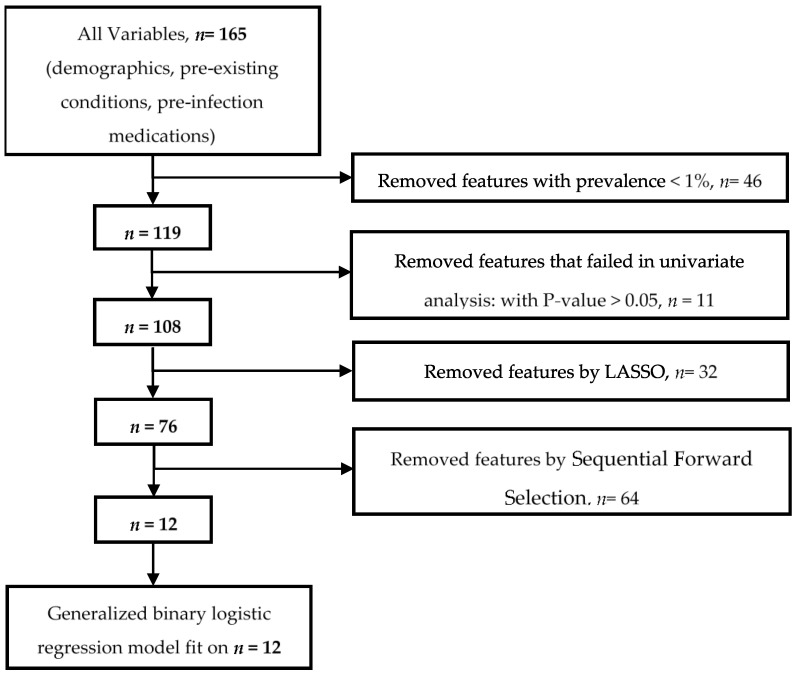
Most important variable selection with a cascade of machine learning methods.

**Table 1 healthcare-10-01244-t001:** Patients’ demographics and characteristics.

Variables	Never Smoker *n* = 380,648 (32.3%)	Former Smoker *n* = 456,348 (38.8%)	Current Smoker *n* = 221,515 (18.8%)	Unknown *n* = 117,795 (10.0%)
Tested positive	85,605 (22.5)	93,665 (20.5)	26,809 (12.1)	24,171 (20.5)
Sex (Male)	318,343 (83.6)	422,136 (92.5)	202,063 (91.2)	748,57 (63.5)
Age [Mean, SD]	59.43 (16.03)	64.77 (14.85)	58.18 (14.19)	50.65 (17.76)
Age 18–30	17,299 (4.5)	10,373 (2.3)	9122 (4.1)	15,584 (13.2)
Age 30–40	42,223 (11.1)	34,708 (7.6)	26,861 (12.1)	26,233 (22.3)
Age 40–50	50,187 (13.2)	35,889 (7.9)	23,508 (10.6)	20,558 (17.5)
Age 50–65	116,777 (30.7)	111,103 (24.3)	83,803 (37.8)	28,990 (24.6)
Age 65–75	101,991 (26.8)	172,289 (37.8)	64,754 (29.2)	15,546 (13.2)
Age 75–85	36,647 (9.6)	67,834 (14.9)	11,668 (5.3)	6539 (5.6)
Age > 85	15,524 (4.1)	24,152 (5.3)	1799 (0.8)	4344 (3.7)
BMI [Mean, SD]	30.74 (6.33)	30.25 (6.33)	28.54 (6.35)	29.01 (5.99)
BMI < 18.5	3048 (0.8)	5820 (1.3)	6732 (3.0)	1919 (1.6)
18.5 ≤ BMI <30	188,058 (49.4)	236,762 (51.9)	133,593 (60.3)	70,140 (59.5)
BMI ≥ 30	189,542 (49.8)	213,766 (46.8)	81,190 (36.7)	45,736 (38.8)
Race				
White	239,184 (62.8)	332,558 (72.9)	150,741 (68.1)	47,987 (40.7)
Black	101,211 (26.6)	83,695 (18.3)	53,037 (23.9)	17,251 (14.6)
Other	40,253 (10.6)	40,095 (8.8)	17,737 (8.0)	52,557 (44.6)
Ethnicity				
Not Hispanic Or Latino	325,720 (85.6)	406,810 (89.1)	202,003 (91.2)	64,420 (54.7)
Hispanic Or Latino	41,593 (10.9)	34,679 (7.6)	12,506 (5.6)	7567 (6.4)
Unknown	13,335 (3.5)	14,859 (3.3)	7006 (3.2)	45,808 (38.9)
**Pre-existing Conditions**				
Acute Cardiac Injury, *n* (%)	6715 (1.8)	13,242 (2.9)	5647 (2.5)	1144 (1.0)
Acute Kidney Failure, *n* (%)	22,825 (6.0)	40,092 (8.8)	16,371 (7.4)	4066 (3.5)
Asthma, *n* (%)	32,606 (8.6)	32,184 (7.1)	11,261 (5.1)	2572 (2.2)
Cancer, *n* (%)	73,409 (19.3)	116,775 (25.6)	48,138 (21.7)	8330 (7.1)
Congestive Heart Failure, *n* (%)	24,805 (6.5)	47,469 (10.4)	15,537 (7.0)	3671 (3.1)
Chronic Lung Disease, *n* (%)	100,324 (26.4)	171,849 (37.7)	92,835 (41.9)	11,099 (9.4)
Chronic Kidney Disease, *n* (%)	49,213 (12.9)	76,070 (16.7)	22,267 (10.1)	5697 (4.8)
Cardiovascular Disease, *n* (%)	113,136 (29.7)	189,991 (41.6)	74,726 (33.7)	13,663 (11.6)
Dementia, *n* (%)	14,937 (3.9)	23,270 (5.1)	6417 (2.9)	4435 (3.8)
Diabetes Any, *n* (%)	119,904 (31.5)	170,917 (37.5)	61,119 (27.6)	12,753 (10.8)
Diabetes Other, *n* (%)	11,692 (3.1)	17,364 (3.8)	6193 (2.8)	1226 (1.0)
Diabetes Type 1, *n* (%)	6767 (1.8)	9445 (2.1)	3445 (1.6)	728 (0.6)
Diabetes Type 2, *n* (%)	118,959 (31.3)	169,769 (37.2)	60,575 (27.3)	12,548 (10.7)
Diabetes With Complications, *n* (%)	83,422 (21.9)	123,064 (27.0)	42,028 (19.0)	8772 (7.4)
Diabetes Without Complications, *n* (%)	108,246 (28.4)	154,275 (33.8)	54,757 (24.7)	10,576 (9.0)
Drug Dependence, *n* (%)	10,998 (2.9)	23,280 (5.1)	34,315 (15.5)	2826 (2.4)
Emphysema, *n* (%)	3092 (0.8)	12,544 (2.7)	8342 (3.8)	597 (0.5)
Heart Disease, *n* (%)	81,153 (21.3)	145,589 (31.9)	52,375 (23.6)	9872 (8.4)
Heart Failure, *n* (%)	31,152 (8.2)	58,606 (12.8)	19,333 (8.7)	4552 (3.9)
Hypertension, *n* (%)	226,370 (59.5)	316,762 (69.4)	132,837 (60.0)	22,791 (19.3)
Hyperlipidemia, *n* (%)	218,820 (57.5)	297,932 (65.3)	121,331 (54.8)	17,705 (15.0)
Lower Respiratory Infection, *n* (%)	32,746 (8.6)	38,228 (8.4)	18,067 (8.2)	4171 (3.5)
Major Depressive Disorder, *n* (%)	138,385 (36.4)	157,528 (34.5)	96,981 (43.8)	14,395 (12.2)
Pneumonia, *n* (%)	15,814 (4.2)	31,011 (6.8)	13,667 (6.2)	3082 (2.6)
Peripheral Artery Disease, *n* (%)	34,538 (9.1)	67,723 (14.8)	31,710 (14.3)	5122 (4.3)
**Pre-infection Medications**				
Metformin, *n* (%)	72,749 (19.1)	100,442 (22.0)	38,267 (17.3)	6021 (5.1)
Non-steroidal Anti-Inflammatory Drug, *n* (%)	235,755 (61.9)	285,628 (62.6)	145,315 (65.6)	27,160 (23.1)
Phenylephrine, *n* (%)	6486 (1.7)	9446 (2.1)	4064 (1.8)	753 (0.6)
Statin, *n* (%)	179,529 (47.2)	264,318 (57.9)	109,726 (49.5)	16,040 (13.6)

**Table 2 healthcare-10-01244-t002:** Patient smoking status before and after imputation.

	Unimputed		Imputed		Gain	Gain, Positive Test
	Total *n* (%)	Tested Positive *n* (%)	Total *n* (%)	Tested Positive *n* (%)		
Never Smoker	380,648 (32.4)	85,605 (37.2)	432,078 (36.7)	97,419 (42.3)	4.3%	5.1%
Former Smoker	456,348 (38.8)	93,665 (40.7)	506,206 (43.0)	103,482 (44.9)	4.2%	4.2%
Current Smoker	221,515 (18.8)	26,809 (11.6)	230,022 (19.6)	29,349 (12.7)	0.8%	1.1%
Unknown	117,795 (10.0)	24,171 (10.5)	-	-		

**Table 3 healthcare-10-01244-t003:** Association between imputed smoking status and COVID-19 positivity for imputed data.

Variables ^†^	Unadjusted		Adjusted	
	*p*-Value	OR (95%CI) ^‡^	*p*-Value	OR (95%CI) ^‡^
Never Smoker		Reference		Reference
Former Smoker	*p* < 0.0001	**0.88 (0.87,0.89)**	*p* < 0.0001	**0.89 (0.88,0.90)**
Current Smoker	*p* < 0.0001	**0.48 (0.47,0.49)**	*p* < 0.0001	**0.51 (0.50,0.52)**
Age ≥ 85			*p* < 0.0001	1.33 (1.30,1.36)
BMI < 18.5			*p* < 0.0001	**0.73 (0.70,0.77)**
BMI ≥ 30			*p* < 0.0001	1.31 (1.30,1.32)
Cancer			*p* < 0.0001	**0.72 (0.71,0.73)**
Dementia			*p* < 0.0001	1.42 (1.39,1.45)
Hispanic or Latino			*p* < 0.0001	1.17 (1.16,1.20)
Low Respiratory Infections			*p* < 0.0001	1.09 (1.07,1.11)
Pneumonia			*p* < 0.0001	1.20 (1.17,1.22)
Phenylephrine			*p* < 0.0001	**0.76 (0.73,0.79)**
Sex (Male)			*p* < 0.0001	1.18 (1.16,1.19)
Septic Shock			*p* < 0.0001	1.07 (1.04,1.10)

^†^ Pre-existing conditions and prehospitalization medications reported for the past 2 years. ^‡^ OR (95%CI) = Odds Ratio and 95% Confidence Intervals. Bold indicates a reduced risk of SARS-CoV-2 infection.

**Table 4 healthcare-10-01244-t004:** Association between imputed smoking status and COVID-19 positivity stratified by age for imputed data.

Variables	Odds Ratio (95%Confidence Intervals)	
	**Unadjusted**	**Adjusted**
Model 1 ^†^		
Former Smoker vs. Never Smoker	**0.88 (0.87,0.89)**	**0.89 (0.88,0.90)**
Current Smoker vs. Never Smoker	**0.48 (0.47,0.49)**	**0.51 (0.50,0.52)**
Model 2 ^‡^		
*Age* ≥ *65*		
Former Smoker vs. Never Smoker	0.99 (0.97, 1.00)	0.98 (0.97, 1.00)
Current Smoker vs. Never Smoker	**0.50 (0.49,0.51)**	**0.52 (0.51,0.53)**
*Age < 65*		
Former Smoker vs. Never Smoker	**0.83 (0.81,0.84)**	**0.83 (0.81,0.84)**
Current Smoker vs. Never Smoker	**0.47 (0.46,0.48)**	**0.48 (0.47,0.49)**

^†^ Model 1 adjusted by Age ≥ 85, BMI < 18.5, BMI ≥ 30, Sex, Former Smoker, Current Smoker, Cancer, Dementia, Hispanic or Latino, Low Respiratory Infections, Pneumonia, Phenylephrine, and Septic Shock. ^‡^ Model 2 stratified by age and adjusted by BMI < 18.5, BMI ≥ 30, Sex, Former Smoker, Current Smoker, Cancer, Dementia, Hispanic or Latino, Low Respiratory Infections, Pneumonia, Phenylephrine and Septic Shock. Bold indicates a reduced risk of SARS-CoV-2 infection.

## Data Availability

The data is available behind the VHA firewall and it cannot leave the VHA electronic health records. Any request for data access requires official approval process.

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
