# Peer review of "Assessing Smoking Status and Risk of SARS-CoV-2 Infection: A Machine Learning Approach among Veterans"

_healthcare, 2022, doi:10.3390/healthcare10071244_

Round 1

Reviewer 1 Report

The work leverages a machine learning approach to characterize the relationship between smoking status and the risk of SARS Covid-2 infection, where a detailed comparison and evidence are provided. However, the machine learning method is not highlighted, and a separate section introducing the machine learning method should be presented. It would be great if different machine learning approaches could be compared. 

Author Response

REVIEWER 1

Q1: “The work leverages a machine learning approach to characterize the relationship between smoking status and the risk of SARS Covid-2 infection, where a detailed comparison and evidence are provided. However, the machine learning method is not highlighted, and a separate section introducing the machine learning method should be presented.”

A1: We appreciate the constructive comment from the reviewer. We revised the method section “2.3.2. The most important variable selection process” to “ 2.3.2. The most important variable selection process using machine learning” (line 116) and added a concise description of each machine learning methods in the text (lines 124-125, lines 129-133, and lines 135-140).

Q2: “It would be great if different machine learning approaches could be compared.”

A2: We agree with the reviewer that different machine learning approaches could be compared. We only used the machine learning methods to sequentially select the most important features that were later included in multivariate logistic regression models and for that reason we decided that adding analysis of different machine learning approaches would not improve the clarity of our manuscript. The approaches we have taken are generally well accepted for reducing the number of potential predictors of an outcome when there are many potential predictors, some of which may be correlated.  

Reviewer 2 Report

The article is well structured and the general scientific backbone is sound. The clarity on the methods is appreciated as is the explanation on how data is processed. The results are exposed quite clearly, not an easy task given the literal wall of numbers the readers face. Overall the article is well presented and the background is explained plainly, both about the pros and the cons of this particular population of patients. Particularly interesting is the application of new technologies such as machine learning to elaborate high volumes of data. Minors grammatical and format errors need a quick revies

Author Response

Q: “The article is well structured and the general scientific backbone is sound. The clarity on the methods is appreciated as is the explanation on how data is processed. The results are exposed quite clearly, not an easy task given the literal wall of numbers the readers face. Overall the article is well presented and the background is explained plainly, both about the pros and the cons of this particular population of patients. Particularly interesting is the application of new technologies such as machine learning to elaborate high volumes of data. Minors grammatical and format errors need a quick revies”

A: We appreciate the positive, scientific comments from the reviewer. We revised to entire manuscript to correct grammatical and format errors (See edits in the whole manuscript).

Reviewer 3 Report

Dear authors, the article is interesting. However, it would be important to revise english language style.

Secondly, i feel that in the introduction there is insufficient evidence supporting the importance of the article.

In the discussion, the authors mention a whole paragraph about other risk factors which is misleading and off topic. The paragraph should be shorter and a paragraph about how smoking could decrease of being positive should be developed with for example some physiology aspects.

The conclusion should be modified too because it does not answer the objective of the study and some phrases are off topic too.

Author Response

Q: “Dear authors, the article is interesting. However, it would be important to revise English language style.”

A: We appreciate the constructive comments from the reviewer. We revised to entire manuscript to improve its readability (See edits in the whole manuscript).

Q: “Secondly, i feel that in the introduction there is insufficient evidence supporting the importance of the article.”

A: We revised the introduction to highlight the importance of the article (lines 55-56).

Q: “In the discussion, the authors mention a whole paragraph about other risk factors which is misleading and off topic. The paragraph should be shorter and a paragraph about how smoking could decrease of being positive should be developed with for example some physiology aspects.”

A: We agree with the reviewer that the paragraph about other risk factors was too long and slightly off topic. In-line with the reviewer’s comment, we revised that section of the discussion (lines 281-293, lines 294-320, lines 322-333, lines 338-346). We also added a paragraph discussing the state of knowledge about the physiology of how smoking could decrease the risk of becoming infected by SARS-CoV-2 (lines 362-382). Finally, we moved lines 346-361 to the Strengths and Weaknesses section of the manuscript

Q: “The conclusion should be modified too because it does not answer the objective of the study and some phrases are off topic too.”

A: We revised the conclusion to align with the objective of the study (lines 430-436).

Reviewer 4 Report

Research work is quite interesting, the authors done extensive literature review and shown good result. The subject addressed in this work is very important and interesting. I just have some suggestions (optional) that can improve the readability of this paper:

·        The author should identify the main findings and justify the novelty and contribution of this work (highlight the significance of your findings).

·        Some paragraphs are too long. Please divide them into several short paragraphs to improve the readability.

·        I would suggest to significatively enrich the description of the proposed techniques, by adding further details. In particular, such description appears to be too superficial and it is not entirely clear. I would suggest to improve such aspect.

·        In the Conclusion section, please explain more about future works

Author Response

Q: “Research work is quite interesting, the authors done extensive literature review and shown good result. The subject addressed in this work is very important and interesting. I just have some suggestions (optional) that can improve the readability of this paper:

The author should identify the main findings and justify the novelty and contribution of this work (highlight the significance of your findings).”

A: We appreciate the positive, scientific comments from the reviewer. We revised the conclusion and highlighted the main findings and the significance of the findings in the discussion.

Q: “Some paragraphs are too long. Please divide them into several short paragraphs to improve the readability.”

A: We revised to entire manuscript and shortened paragraphs in the discussion to improve its readability (See edits in the whole manuscript).

Q: “I would suggest to significatively enrich the description of the proposed techniques, by adding further details. In particular, such description appears to be too superficial and it is not entirely clear. I would suggest to improve such aspect.”

A: We revised the method section “2.3.2. The most important variable selection process” to “2.3.2. Identifying the most important variables using a machine learning selection process” and added concise description of each machine learning methods in the text.

Q: “In the Conclusion section, please explain more about future works”

A: We revised the conclusion to provide more details for future works (lines 434-436). 

Round 2

Reviewer 1 Report

The authors resolved most of my concerns about the manuscript, I believe the overall quality of the current version is good. 

As for the attempt of the machine learning methods, I suggest the authors should cite more of the related papers like the following one:

Qi, Jun, and Javier Tejedor. "Deep multi-view representation learning for multi-modal features of the schizophrenia and schizo-affective disorder." In 2016 IEEE International Conference on Acoustics, Speech and Signal Processing (ICASSP), pp. 952-956. IEEE, 2016.

After the revision of this round, I believe the manuscript should be published. 

Author Response

Q1: The authors resolved most of my concerns about the manuscript, I believe the overall quality of the current version is good. 

A1: We appreciate the positive, scientific comments from the reviewer. We further revised to entire manuscript to correct grammatical and format errors (See edits in the whole manuscript).

Q2: As for the attempt of the machine learning methods, I suggest the authors should cite more of the related papers like the following one:

Qi, Jun, and Javier Tejedor. "Deep multi-view representation learning for multi-modal features of the schizophrenia and schizo-affective disorder." In 2016 IEEE International Conference on Acoustics, Speech and Signal Processing (ICASSP), pp. 952-956. IEEE, 2016.

A2: We agree with the reviewer that more machine learning related papers should be cited. We revised the Strengths and Weaknesses section and cited the paper by Qi et al. 2016 (lines 384-386. We also added a recent review paper of machine learning methods used to predict hospital admission by Huang et al. 2021 (lines 386-389).

Q3: After the revision of this round, I believe the manuscript should be published. 

A3: We appreciate the constructive comment from the reviewer.